# Originalities of Willow of *Salix atrocinerea* Brot. in Mediterranean Europe

**Mauro Raposo** [1,*], **Ricardo Quinto-Canas** [2,3], **Ana Cano-Ortiz** [4], **Giovanni Spampinato** [5] **and Carlos Pinto Gomes** [1]

1   Department of Landscape, Environment and Planning, Mediterranean Institute for Agriculture, Environment and Development (MED), School of Science and Technology—Universidade de Évora, Rua Romão Ramalho, n° 59, P-7000-671 Évora, Portugal; cpgomes@uevora.pt
2   Faculty of Sciences and Technology, Campus de Gambelas, University of Algarve, P-8005-139 Faro, Portugal; rjcanas@ualg.pt
3   CCMAR—Centre of Marine Sciences (CCMAR), Campus de Gambelas, University of Algarve, P-8005-139 Faro, Portugal
4   Department of Animal and Plant Biology and Ecology, Section of Botany, University of Jaén, Campus Universitario Las Lagunillas s/n., 23071 Jaén, Spain; anacanor@hotmail.com
5   Department of Agraria, "Mediterranea" University of Reggio Calabria, Loc. Feo di Vito, 89122 Reggio Calabria, Italy; gspampinato@unirc.it
*   Correspondence: mraposo@uevora.pt

**Abstract:** Willow communities (genus *Salix*) occurring in Mediterranean Europe are presented, showing, through statistical treatment with multivariate cluster analysis, the separation of the different plant communities and their sintaxonomic affiliation. Six willow communities have been identified, whose formations include a set of plants with high heritage value. We highlight plants with legal protection status (Annex IV and II of the Habitats Directive-92/43/EEC), endemic, rare, and endangered species such as *Salix salviifolia* subsp. *australis*, *Cheirolophus uliginosus*, *Euphorbia uliginosa* and *Leuzea longifolia*. Therefore, two new willow communities are proposed for the southwest of the Iberian Peninsula. The first dominated by *Salix atrocinerea*, *Frangulo baticae-Salicetum atrocinereae ass. nova* of ribatagan distribution, under acid substrates, thermomediterranean to lower mesomediterranean, dry to sub-humid. The second, dominated by the endemic *Salix salviifolia* subsp. *australis*, *Clematis flammulae-Salicetum australis* distributed in the Algarve, developing on neutral-basic substrates, exclusively thermomediterranous, dry to sub-humid. In both cases, there are presented on their own floristic serial, ecology, and substitution steps. A new hygrophytic meadows was also identified dominated by *Molinia caerulea* subsp. *arundinaceae*, *Cheirolopho uliginosii-Molinietum arundinaceae ass. new hoc loco*, which lives on substrates rich in organic matter, exclusive to the Ribatagano Sector. Through the deepening of knowledge about the composition and dynamics of riparian vegetation, it is possible to adapt management methods to sustain and protect these important edafo-hygrophilic systems in the Mediterranean.

**Keywords:** cluster analysis; geobotany; peatland; phytosociology; willow forest; Sardinia; southwest of Iberian Peninsula

## 1. Introduction

Riparian zones are highly heterogeneous and disturbed environments. They are composed of a wide variety of physical habitats in terms of their size of substrate sediment, moisture, and nutrient conditions, inundation duration and frequency, and also susceptibility to drought [1]. The southern

Iberian small rivers and torrential streams are characterized by extreme flow irregularity (floods and droughts), leading to decreased soil cohesion and high eroded sediment discharge.

As a consequence, it is necessary to increase knowledge on Mediterranean riverside vegetation, in order to contribute to select methods of management and conservation of these highly important systems, especially in territories with a torrential character [2]. This importance is related to ecosystem services, guaranteeing environmental sustainability, through the regulation of the hydrological cycle, erosion control, and refuge of a large number of floristic and fauna species. A territory to be successful at the environmental, social, and economic level must have a balanced landscape [3]. The willow forests comprise the potential natural vegetation in torrential intermittent streams of the Southwestern part of the Iberian Peninsula [4].

According to Rivas-Martínez [5], Costa et al. [6] and Mucina et al. [7], two vegetation classes of riparian woodlands can be recognized in southwestern Europe: *Alnetea glutinosae* and *Salici purpureae-Populetea nigrae*. The first encompasses swamp and fens forests of *Alnus glutinosa* and *Salix atrocinerea*, in margins frequently inundated by dystrophic lentic waters of the Coastal Lusitania and West Andalusia Province [5,6], which are included in the *Salici atrocinereae-Alnenion glutinosae* suballiance (*Alnion glutinosae* alliance). The second includes willow communities of *Salix salviifolia* subsp. *australis* linked to the *Salicion salviifoliae* alliance, found in riparian, hygrophilous, deciduous forests growing on alluvial soils in the Mediterranean, sub-Mediterranean, and Thermo Atlantic Regions.

This paper provides a phytosociological and syntaxonomical analysis of the riparian woodlands communities dominated by *Salix salviifolia* subsp. *australis* (included in the *Salicion salviifoliae* alliance) and *Salix atrocinerea* (included in the *Alnion glutinosae* alliance), which occur in the southwest of the Iberian Peninsula and in Sardinia. In Iberian Peninsula *Salix atrocinerea* grows in some in riparian communities or in humid depressions that fall within the *Salici atrocinereae-Alnenion glutinosae* sub-alliance of the *Alnion glutinosae* [5]. Costa et al. [6] consider it characteristic *Carici lusitanicae-Salicetum atrocinereae*. This association is exclusive to the Sado river basin [8]. It is also recognized as *Viti sylvestris-Salicetum atrocinereae* occurring from the south of Spain [9] to the center of Portugal [4].

The communities of *Salix salviifolia* subsp. *australis* fall within *Salicion salviifoliae*, represented by the association *Salicetum atrocinereae-australis*, to the Alentejo territories [2].

In Sardinia, *Salix atrocinerea* grows in some in riparian communities or in humid depressions that fall within the *Hyperico hircini-Alnenion glutinosae* sub-alliance of the *Osmundo Alnion* [10]. In particular, Angius and Bacchetta [11] consider it characteristic of *Carici microcarpae-Salicetum atrocinereae* willow woodlands. Biondi and Bagella [12] described the *Myrto communis-Salicetum atrocinereae* for the marshy area of the island of Maddalena (Nord East Sardinia). As a subordinate species, it is also found in other riparian plant communities such as of the riparian woods of the *Eupatorio corsici-Alnetum glutinosae* were differentiate the sub-association *salicetosum atrocinereae*.

## 2. Materials and Methods

The sampling was carried out according to the school of Zürich-Montpellier (also known as Sigmatist) approach [13–18]. Taxonomic nomenclature followed the work of Coutinho [19], Franco [20,21], Franco and Rocha Afonso [22], Castroviejo et al. [23], and Valdés et al. [24]. The identification of *Salix* genus followed Portela-Pereira et al. [25]. Seventy-one relevés collected from bibliography along with thirteen relevés carried out by us were submitted to hierarchical cluster analysis using Ward's method with Euclidean distance to measure dissimilarity [26], using the software R. The transformation of cover-abundance values follows Van der Maarel [27]. The relevés performed in data matrix include our field sampling and relevés taken from literature: Costa [2], Cano and Valle [28], Rivas-Martínez [9], Biondi and Bagella [12], and Angius and Bacchetta [11]. For biogeographical and bioclimatological information we follow Rivas-Martínez [29], Rivas-Martínez et al. [30,31], and Fenu et al. [32].

*Data Collection*

For the confirmation of the new associations we carry out a set of relevés (Figure 1). In each relevé, all existing plants were recorded and assigned a quantitative index according to the abundance-dominance scale, with according by Braun-Blanquet [33]. This scale combines an estimative between the number of individuals of each existing specie and the respective area within the inventory. For each index (in bold), there is a coverage range, namely: + few individuals with very poor coverage (0.1% to 1%); 1 very abundant individuals with low coverage (from 1% to 10%); 2 individuals very abundant or covering at least 1/20 of the surface (from 10% to 25%); 3 any number of individuals covering $\frac{1}{4}$ to $\frac{1}{2}$ of the surface (from 25% to 50%); 4 any number of individuals covering $\frac{1}{2}$ to $\frac{3}{4}$ of the surface (from 50% to 75%); 5 any number of individuals covering more than $\frac{3}{4}$ of the surface (from 75% to 100%). We compiled an Excel© table with 84 relevés, where 13 were made by the authors and the rest supported by bibliography [2,9,11,12].

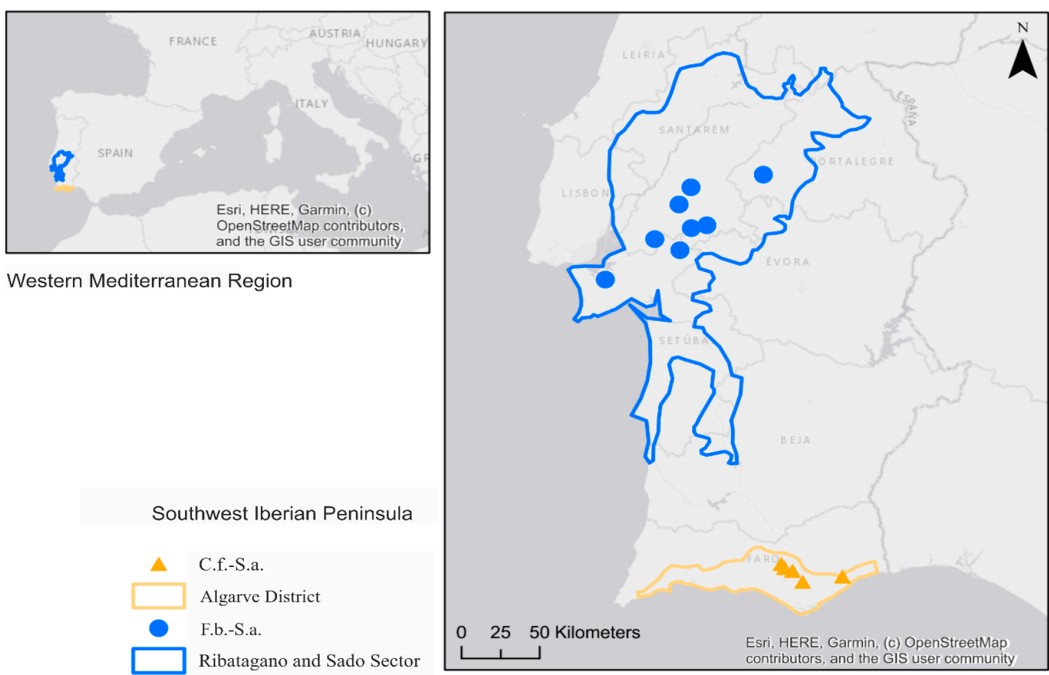

**Figure 1.** Sampling sites in the southwest of the Iberian Peninsula.

## 3. Results and Discussion

The cluster analysis (Figure 2) of 84 riparian relevés result in two main clusters with high dissimilarity levels, which are organized into two different groups (Appendix A. Syntaxonomical Scheme). Group A has a high dissimilarity in relation to the other associations, and includes communities dominated by *Salix salviifolia* subsp. *australis* (*Salicion salviifoliae* alliance), which spreads across the southwestern areas of the Iberian Peninsula: (Subgroup A1) *Clematido flammulae-Salicetum australis* (plot 14–18) and (Subgroup A2) *Salicetum atrocinereo-australis* (plot 1–11). The samples ascribed to the communities dominated by *Salix atrocinerea*, occurring throughout the southwest of the Iberian Peninsula and the island of Sardinia, are included in the cluster Group B, divided into two subgroups. The relevé cluster subgroup B1 corresponds to the association *Myrto communis-Salicetum atrocinerea* (plot 39–46), which occurs in thermomediterranean bioclimatic areas, mainly of the islands Caprera and Santo Stefano. The subgroup B2 comprises two block communities: The first block (B21) encompasses plot (54–84) contained within the association *Carici microcarpae-Salicetum atrocinereae* described for the thermomediterranean to mesomediterranean, dry to sub-humid belts of the island of Sardinia, which are included in the *Salici purpureae-Populetea nigrae* class. The second block (B22) encompasses samples ascribed to the associations included in the *Alnetea glutinosae* vegetation class: *Carici lusitanicae-Salicetum*

*atrocinereae* (plot 27–38), suggested by Neto [8] for the Sadese District; *Viti sylvestris-Salicetum atrocinerea* (plot 47–53), described by Rivas-Martínez [9] for the thermomediterranean belt of the Cádiz and Litoral Huelva Sector, extending over the Coastal Lusitania and West Andalusia Province, and; *Frangulo baeticae-Salicetum atrocinereae* (plot 19–26), which is proposed here as a new association for the Ribatejo and Sado Sector. In Table 1, the characteristics and differentials species of each association studied is highlighted.

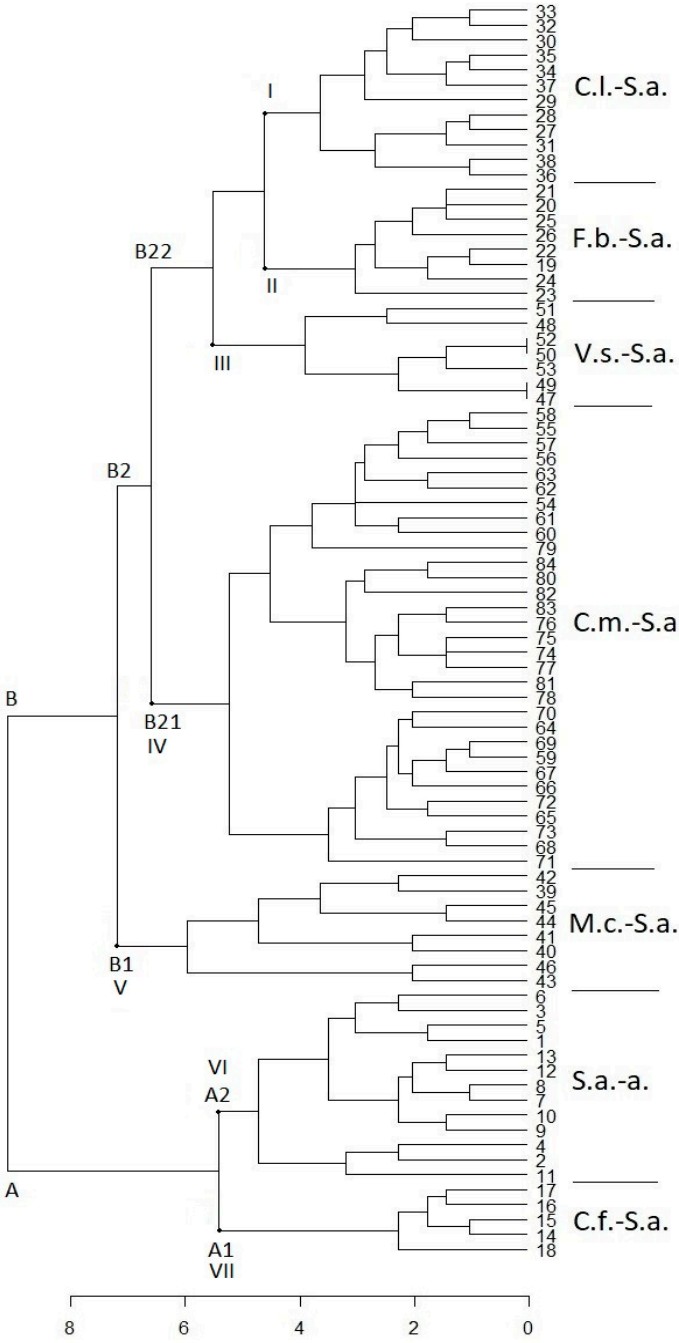

**Figure 2.** Dendrogram with the communities dominated by *Salix atrocinerea* and *Salix salviifolia* subsp. *australis*, occurring throughout the southwest of the Iberian Peninsula and the island of Sardinia. B22 (27–38)—C.l.-S.a.: *Carici lusitanicae-Salicetum atrocinereae*; B22 (19–26)—F.b.-S.a.: *Frangulo baeticae-Salicetum atrocinereae ass. nova hoc. loco*; B22 (47–53)—v.s.-S.a.: *Viti sylvestris-Salicetum atrocinereae*; B21 (54–84)—C.m.-S.a.: *Carici microcarpae-Salicetum atrocinerea*; B1 (39–46)—M.c.-S.a.: *Myrto communis-Salicetum atrocinerea*; A2 (1–13)—S.a.-a.: *Salicetum atrocinereo-australis*; A1 (14–18)—C.f.-S.a.: *Clematido flammulae-Salicetum australis ass. nova hoc. loco*.

**Table 1.** Synoptic table with the communities of *Salix atrocinerea* and *Salix salviifolia* subsp. *australis* in south-west Europe.

| Numerical Order | A | B | C | D | E | F | G |
|---|---|---|---|---|---|---|---|
| Number of relevées | 12 | 8 | 7 | 31 | 8 | 13 | 5 |
| Acronym | C.l.-S.a. | F.b.-S.a. | V.s.-S.a. | C.m.-S.a. | M.c.-S.a. | S.a.-a. | C.f.-S.a. |
| **Characteristics and differentials** | | | | | | | |
| *Salix atrocinerea* | V | V | V | V | V | IV | . |
| *Carex lusitanica* | V | . | . | . | . | . | . |
| *Thelypteris palustris* | V | . | V | . | . | . | . |
| *Myrica gale* | V | . | . | . | . | . | . |
| *Frangula alnus* subsp. *baetica* | IV | V | . | . | . | . | . |
| *Salix salviifolia* subsp. *australis* | II | II | . | . | . | V | V |
| *Cheirolophus uliginosus* | . | III | . | . | . | . | . |
| *Leuzea longifolia* | . | II | . | . | . | . | . |
| *Euphorbia uliginosa* | . | II | . | . | . | . | . |
| *Vitis sylvestris* | . | . | V | . | . | III | . |
| *Rubus ulmifolius* | . | . | IV | V | V | V | . |
| *Fraxinus angustifolia* | . | . | II | . | . | II | III |
| *Dioscorea communis* | . | . | . | V | . | II | . |
| *Carex microcarpa* | . | . | . | V | . | . | . |
| *Smilax aspera* | . | . | . | V | . | . | . |
| *Quercus ilex* | . | . | . | V | . | . | . |
| *Brachypodium sylvaticum* | . | . | . | IV | . | . | II |
| *Oenanthe crocata* | . | . | . | III | IV | . | . |
| *Hypericum hircinum* subsp. *hircinum* | . | . | . | III | . | . | . |
| *Euphorbia meuselii* | . | . | . | III | . | . | . |
| *Nerium oleander* | . | . | . | III | . | . | . |
| *Bellium bellidioides* | . | . | . | III | . | . | . |
| *Bryonia dioica* | . | . | . | . | . | II | . |
| *Securinega tinctorea* | . | . | . | . | . | + | . |
| *Salix neotricha* | . | . | . | . | . | III | V |
| *Clematis flammula* | . | . | . | . | . | . | V |
| *Vinca difformis* | . | . | . | . | . | . | IV |
| *Equisetum ramosissimum* | . | . | . | . | . | . | IV |
| *Aristolochia paucinervis* | . | . | . | . | . | . | I |
| *Equisetum telmateia* | . | . | . | . | . | . | I |
| *Hedera canariensis* | . | . | . | . | . | II | . |
| *Myrtus communis* | . | . | . | . | V | . | . |
| *Carex hispida* | . | . | . | . | III | . | . |
| *Tamarix africana* | . | . | . | . | III | . | . |

Other taxa: + Alnus lusitanica, + Populus alba in A; + Arum italicum, + Vincetoxum nigrum in C; + Populus nigra, + Securigena tinctoria in F. Note: A—Carici lusitanicae-Salicetum atrocinereae; B—Frangulo baeticae-Salicetum atrocinereae ass. nova hoc. loco; C—Viti sylvestris-Salicetum atrocinereae; D—Carici microcarpae-Salicetum atrocinerea; E—Myrto communis-Salicetum atrocinerea; F—Salicetum atrocinereo-australis; G—Clematido flammulae-Salicetum australis ass. nova hoc. loco.

*Description of Willow Communities*

I—*Carici lusitanicae-Salicetum atrocinereae* (clusters 27–38, Figure 1)

According to Neto [8], this association is dominated by *Salix atrocinerea* and occupies swamps, lagoons, and margins of rivers and streams with water flowing over sandy soils subjected to fluctuations in its flow (although they present a high humidity content during the dry season). This association tends to be located in hygrophilous environments in a dry to subhumid ombroclimate in the thermomediterranean belt of the Sadese District.

II—*Frangulo baeticae-Salicetum atrocinereae ass. nova hoc. loco* (clusters 19–26, Figure 1; holotypus relevé: 08, Table 2)

**Table 2.** *Frangulo baeticae-Salicetum atrocinereae ass. nova hoc. loco* (*Salici atrocinereae-Alnenion glutinosae, Alnion glutinosae, Alnetalia glutinosae, Alnetea glutinosae*).

| Ordinal Number | 1 | 2 | 3 | 4 | 5 | 6 | 7 | 8 |
|---|---|---|---|---|---|---|---|---|
| Surface (m$^2$) | 150 | 100 | 150 | 300 | 200 | 250 | 200 | 100 |
| Altitude (m) | 48 | 64 | 43 | 86 | 65 | 27 | 45 | 50 |
| Cover rate (%) | 85 | 80 | 80 | 95 | 85 | 80 | 85 | 90 |
| Orientation | W | W | N | S | NW | W | N | N |
| Slope (%) | 12 | 18 | 12 | 6 | 10 | 8 | 7 | 15 |
| Average heigth (m) | 4 | 5 | 4 | 4 | 5 | 3 | 4 | 3 |
| Number of species | 24 | 22 | 23 | 22 | 28 | 32 | 36 | 36 |
| Association characteristics | | | | | | | | |
| *Salix atrocinerea* | 4 | 3 | 4 | 4 | 3 | 4 | 4 | 4 |
| *Frangula alnus* subsp. *baetica* | + | 1 | . | 2 | 1 | 1 | + | 3 |
| *Cheirolophus uliginosus* | . | . | + | . | 1 | . | 2 | + |
| *Leuzea longifolia* | . | . | . | + | + | 1 | . | . |
| *Salix salviifolia* subsp. *australis* | . | . | . | . | . | + | . | + |
| *Euphorbia uliginosa* | . | . | . | . | . | . | + | + |
| Companions | | | | | | | | |
| *Molinia caerulea* subsp. *arundinacea* | + | 1 | 1 | 1 | 2 | 2 | 3 | 3 |
| *Rubus ulmifolius* | + | + | 1 | + | + | + | + | + |
| *Quercus suber* | + | + | + | + | + | + | + | + |
| *Erica ciliaris* | 1 | . | 2 | 1 | 1 | 1 | 1 | 1 |
| *Ulex australis* subsp. *welwitschianus* | + | + | + | + | + | . | + | + |
| *Brachypodium phoenicoides* | 1 | + | . | 1 | 1 | + | + | 1 |
| *Ulex minor* var. *lusitanicus* | . | + | 1 | 1 | 1 | 1 | 1 | 1 |
| *Erica erigena* | 1 | + | 1 | . | . | 1 | 1 | 2 |
| *Lonicera hispanica* | . | + | 1 | . | 2 | 1 | + | + |
| *Schoenus nigricans* | . | . | + | + | 1 | 1 | + | 1 |
| *Calluna vulgaris* | + | + | . | . | + | + | + | + |
| *Asparagus aphyllus* | + | + | . | + | + | . | + | + |
| *Schirpus holoschoenus* | + | + | + | . | . | + | + | + |
| *Daphne gnidium* | + | . | . | + | + | + | . | + |
| *Agrostis castellana* | + | + | . | + | . | + | + | . |
| *Pteridium aquilinum* | . | . | 1 | + | . | + | + | + |
| *Lepidophorum repandum* | . | . | . | + | + | + | + | + |
| *Prunella vulgaris* | . | + | . | + | + | + | + | . |
| *Holcus lanatus* | + | + | + | . | . | . | + | + |
| *Juncus rugosus* | + | . | + | + | . | . | . | + |
| *Crataegus monogyna* | . | . | + | + | + | + | . | . |
| *Erica scoparia* | + | . | + | . | . | . | + | + |
| *Panicum repens* | + | + | + | . | . | . | + | . |
| *Quercus lusitanica* | . | + | . | . | + | . | + | + |
| *Juncus effusus* | 1 | + | . | . | . | + | . | . |
| *Erica arborea* | . | . | . | . | + | + | . | + |
| *Arbutus unedo* | . | . | . | . | + | + | + | . |
| *Pseudarrhenatherum longifolium* | . | . | . | . | + | + | . | + |
| *Genista triacanthos* | + | . | . | . | + | . | . | + |
| *Euphorbia transtagana* | . | . | + | . | . | . | + | + |
| *Agrostis stolonifera* | . | . | + | . | + | + | . | . |
| *Halimium calycinum* | . | + | . | . | + | . | . | + |
| *Fuirena pubescens* | . | + | + | + | . | . | . | . |
| *Stachys officinalis* | . | . | . | . | + | + | + | . |
| *Halimium lasianthum* | + | . | . | . | . | . | + | + |
| *Erica lusitanica* | . | . | 1 | . | . | + | . | . |
| *Lavandula lusitanica* | . | . | . | . | + | . | . | + |
| *Danthonia decumbens* | . | . | . | . | + | . | + | + |
| *Narcissus bulbocodium* | . | . | . | . | . | + | . | + |
| *Holcus mollis* | + | . | . | . | . | . | . | + |
| *Hyacinthoides transtagana* | . | . | . | . | . | + | . | + |

**Table 2.** *Cont.*

| Ordinal Number | 1 | 2 | 3 | 4 | 5 | 6 | 7 | 8 |
|---|---|---|---|---|---|---|---|---|
| *Laurus nobilis* | + | . | . | . | . | . | + | . |
| *Lobela urens* | . | . | . | . | . | + | + | . |
| *Potentilla erecta* | . | . | . | . | . | + | + | . |

Other taxa: + Myrtus communis in 1; Dactylis hispanica subsp. lusitanica in 2; + Carex riparia in 4; + Carex flacca in 5; + Cyperus longus, + Juncus inflexus, + Juncus conglomeratus in 7; + Lavandula luisieri in 8. Localities (Coordinate Reference System Datum WGS84): 1—Fajarda (lat 38°58′30.8″ N, long 8°36′47.3″ W); 2—Malhada Alta (lat 38°52′00.2″ N, long 8°32′36.9″ W); 3—Canha (lat 38°45′01.1″ N, long 8°34′28.2″ W); 4—Montargil (lat 39°08′13.9″ N, long 8°08′11.8″ W); 5—Foros do Rebocho (Monte da Barca; lat 38°52′48.6″ N, long 8°28′39.6″ W); 6—Infantado (lat 38°50′34.6″ N, long 8°44′07.5″ W); 7—Coina (lat 38°36′43.4″ N, long 9°02′21.2″ W); 8—Venda (lat 39°03′43.4″ N, long 8°32′50.8″ W).

The *Frangulo baeticae-Salicetum atrocinereae ass. nova hoc. loco* occurs in thermomediterranean and warmer mesomediterranean areas, dry to sub-humid belts of the Ribatejo and Sado Sector, over hydromorphic sandy soils rich in organic matter, with some degree of peat formation, and are found on unusual position, since these willow woodlands are strictly linked to slopes of hills with a supplementary supply of springs. This is because the surface water infiltrates through the permeable pliocene-pleistocene sandy soils and feed the upwelling of subterranean water in many isolated springs, often in positions where the impermeable miocene substrates emerge. Thus, *Frangulo baeticae-Salicetum atrocinereae* is an silicicolous association developed close to springs, supporting a long period of waterlogging and represents an obvious discontinuities (island) with the surrounding zonal forests of cork oak (*Quercus suber*): *Aro neglecti–Querco suberis sigmetum*, which represent the climatophilous vegetation series. In addition to their distinct ecology, this new acidophilous community are physiognomically characterized by vegetation that is markedly different from the willow forests most often associated with the banks of water courses of the valleys from the nearby areas. The new *Salix atrocinerea* willow forest proposed here is frequently accompanied by *Frangula alnus* subsp. *baetica* and other differential species from the *Genistion micrantho-anglicae* alliance, such as *Cheirolophus uliginosus*, *Leuzea longifolia* and *Euphorbia uliginosa*.

Sinecologically, these willow woodlands fit in the *Salici atrocinereae-Alnenion glutinosae* sub-alliance (*Alnion glutinosae, Alnetalia glutinosae, Alnetea glutinosae*) and represent the mature stage of the edaphohygrophilous series: *Frangulo baeticae-Salico atrocinereae sigmentum*. The *Cirsio welwitschii-Ericetum ciliaris*, a hydromorphic heathland dominated by *Erica ciliaris* and *Ulex minor* var. *lusitanicus*, represents the first seral stage.

With the destruction of these hygrophyllous formation occurs a new association dominated by *Molinia caerulea* subsp. *arundinaca*, accompaneid by *Cheirolophus uliginosus*: *Cheirolopho uliginosii-Molinietum arundinaceae ass. nova hoc loco* (*holotypus* relevé: 07, Table 3). This perennial grassland grows on peaty areas of the Ribatejo and Sado Sector, colonizing sandy soils with a water-table near the surface. We include the phytocoenoses within the *Brizo minoris-Holoschoenenion vulgaris* sub-alliance (*Molinio arundinacea-Holoschoenion vulgaris, Holoschoenetalia vulgaris, Molinio-Arrhenatheretea*), which also differs from the other association already described by a pool of rare or endemic species, such as *Cheirolophus uliginosus*, *Leuzea longifolia* and *Euphorbia uliginosa*.

**Table 3.** Cheirolopho uliginosii-Molinietum arundinaceae ass. nova hoc loco (Brizo minoris-Holoschoenenion vulgaris, Molinio arundinacea-Holoschoenion vulgaris, Holoschoenetalia vulgaris, Molinio-Arrhenatheretea).

| Ordinal Number | 1 | 2 | 3 | 4 | 5 | 6 | 7 |
|---|---|---|---|---|---|---|---|
| Surface (m$^2$) | 30 | 25 | 20 | 20 | 20 | 20 | 25 |
| Altitude (m) | 84 | 49 | 26 | 66 | 61 | 42 | 46 |
| Cover rate (%) | 95 | 95 | 85 | 80 | 85 | 90 | 95 |
| Orientation | S | N | W | NW | NE | N | N |

**Table 3.** *Cont.*

| Ordinal Number | 1 | 2 | 3 | 4 | 5 | 6 | 7 |
|---|---|---|---|---|---|---|---|
| Slope (%) | 6 | 12 | 5 | 3 | 8 | 10 | 8 |
| Average heigth (m) | 0.7 | 0.8 | 0.7 | 0.6 | 0.7 | 0.6 | 0.8 |
| Number of species | 25 | 26 | 33 | 21 | 26 | 27 | 31 |
| Association characteristics | | | | | | | |
| *Molinia caerulea* subsp. *arundinacea* | 5 | 5 | 5 | 4 | 5 | 5 | 5 |
| *Schoenus nigricans* | 1 | 2 | 2 | + | + | 1 | + |
| *Schirpus holoschoenus* | + | + | + | + | + | + | + |
| *Cheirolophus uliginosus* | + | . | + | 1 | + | + | 2 |
| *Prunella vulgaris* | + | . | + | . | + | + | + |
| *Erica erigena* | . | + | + | + | . | + | 1 |
| *Holcus lanatus* | . | + | . | + | + | 1 | + |
| *Euphorbia uliginosa* | . | + | . | . | . | . | + |
| *Leuzea longifolia* | . | . | + | + | . | . | . |
| *Fuirena pubescens* | . | . | . | . | + | + | . |
| Companions | | | | | | | |
| *Erica ciliaris* | 1 | 1 | + | 1 | + | + | + |
| *Ulex minor* var. *lusitanicus* | + | 1 | + | + | + | 1 | + |
| *Brachypodium phoenicoides* | + | + | 1 | + | + | + | + |
| *Salix atrocinerea* | + | + | + | + | + | + | . |
| *Ulex australis* subsp. *welwitschianus* | . | + | + | + | + | + | + |
| *Pteridium aquilinum* | 1 | . | . | + | 2 | + | + |
| *Calluna vulgaris* | + | + | + | + | . | . | 1 |
| *Daphne gnidium* | + | + | + | . | . | + | + |
| *Holcus mollis* | + | + | + | . | + | . | + |
| *Erica scoparia* | . | + | + | . | + | + | + |
| *Euphorbia transtagana* | + | . | + | . | . | + | 1 |
| *Crataegus monogyna* | + | + | . | + | + | . | . |
| *Agrostis castellana* | + | . | . | . | + | + | + |
| *Asparagus aphyllus* | + | + | + | . | . | . | + |
| *Lepidophorum repandum* | . | + | + | . | + | . | + |
| *Juncus rugosus* | . | . | + | . | + | + | . |
| *Ditrichia viscosa* | + | . | + | . | + | . | . |
| *Juncus effusus* | + | . | . | . | . | + | + |
| *Potentilla erecta* | + | . | + | . | . | . | + |
| *Frangula alnus* subsp. *baetica* | + | + | + | . | . | . | . |
| *Halimium calycinum* | . | + | + | . | + | . | . |
| *Stachys officinalis* | . | . | + | + | . | . | + |
| *Halimium lasianthum* | . | . | + | . | . | + | + |
| *Lobelia urens* | . | . | + | + | . | . | + |
| *Hyacinthoides transtagana* | . | . | + | + | . | . | + |
| *Cistus psilosepalus* | . | + | . | . | . | + | + |
| *Panicum repens* | . | . | . | . | + | + | + |
| *Erica lusitanica* | + | . | . | . | 1 | . | . |
| *Narcissus bulbocodium* | + | . | + | . | . | . | . |
| *Lonicera hispanica* | + | + | . | . | . | . | . |
| *Erica arborea* | . | . | + | . | . | + | . |
| *Carex riparia* | . | . | . | 1 | . | + | . |
| *Agrostis stolonifera* | . | + | . | . | + | . | . |
| *Euphorbia boetica* | . | + | . | . | + | . | . |

Other taxa: + Salix salviifolia subsp. australis in 1; + Dactylis hispanica subsp. lusitanica; + Hypericum perforatum in 2; + Carex lusitanica; + Pinguicula lusitanica, + Scutellaria minor; in 3; + Hypericum elodes, + Myrtus communis in 4; + Agrostis juressi in 5; + Asphodelus aestivus; + Cynodon dactylon in 6; + Danthonia decumbens; + Potentilla erecta in 7. Localities (Coordinate Reference System Datum WGS84): 1—Montargil (lat 39°07′29.2″ N, long 8°07′45.8″ W); 2—Venda (lat 39°03′42.7″ N, long 8°32′51.7″ W); 3—Samora Correia (Infantado, lat 38°49′11.8″ N, long 8°45′24.8″ W); 4—Foros do Rebocho (Monte da Barca; lat 38°52′34.0″ N, long 8°27′43.5″ W); 5—Coruche (lat 38°52′43.7″ N, long 8°28′42.5″ W); 6—Canha (lat 38°45′01.4″ N, long 8°34′26.7″ W); 7—Palhais (Mata da Machada; lat 38°36′43.4″ N, long 9°01′56.0″ W).

III—*Viti sylvestris-Salicetum atrocinereae* (clusters 47–53, Figure 1)

This is a community that was described as belonging to the gleyed sandy soils of the Doñana area, in Cádiz and Littoral Huelva Sector [9], extending towards the Coastal Lusitania and West Andalusia Province, and reaching the Atlantic Orolusitania Subprovince and western areas of the Lusitania and Extremadura Subprovince. It occurs in dry to sub-humid thermomediterranean belt, and grows on ologotrophic soils, subjected to temporary flooding. According to Rivas-Martínez [9], this community is dominated by *Salix atrocinerea* and is characterized by the frequency of *Vitis vinifera* subsp. *sylvestris*, *Thelypteris palustris*, among others.

IV—*Carici microcarpae-Salicetum atrocinereae* (clusters 54–84, Figure 1)

The Carici microcarpae-Salicetum atrocinereae, linked to the Hyperico hircini-Alnenion glutinosae suballiance (Osmundo-Alnion, Populetalia albae, Salici purpureae-Populetea nigrae), encompasses willow forests dominated by Salix atrocinerea usually accompanied by Carex microcarpa, supporting longer flooding periods of both oligotrophic and oligo-eutrophic waters, and widely distributed throughout the thermomediterranean to mesomediterranean, dry to sub-humid belts of the Sardinian–Corsican province [32]. According to Angius and Bacchetta [11], this association has a wide representation of endemic species characteristics of Hyperico hircini-Alnenion glutinosae suballiance, such as Mentha suaveolens subsp. insularis, Eupatorium cannabium subsp. corsicum, Euphorbia meuselii.

V—*Myrto communis-Salicetum atrocinerea* (clusters 39–46, Figure 1)

For La Madalena archipelago, located between north-eastern Sardinia and southern Corsica (Maddalenino Subscector, Campidanese-Turritano Sector, Sardinian–Corsican province), Biondi and Bagella [12] published the willow forest *Myrto communis-Salicetum atrocinerea*, dominated by *Salix atrocinerea*, often accompanied by *Rubus ulmifolius*, *Carex hispida*, *Myrtus communis* and *Oenanthe crocata*. This association is located in hydromorphic soils on swampy depressions, where the water table is almost permanent or close to the soil surface, within the thermomediterranean dry bioclimatic belt. On more dry soils of riverbeds, occurs a variant of this association, the subass. *tamaricetosum africanae*, characterized by the presence of *Tamarix africana* [12]. Following Bacchetta [34], in river and streams with permanent flow, on oligo-miocene deposits located in upper thermomediterranean or lower mesomediterranean belts under upper dry to lower subhumid areas of the western-central Sardinia, the communities enriched with *Laurus nobilis* are classified as subass. *lauretosum nobilis*.

VI—*Salicetum atrocinereo-australis* (clusters 1–13, Figure 1)

This association colonizes siliceous soils of the thermomediterranean to mesomediterranean, dry to humid belts of the southwestern part of the Cádiz and Sado Suprovince and Lusitania and Extremadura Subprovince [2,4]. According to Dalila [35] and Quinto-Canas [36], the *Salicetum atrocinereo-australis* occurs in periodically flooded margins of temporary watercourses, characterized by torrential flows during the wet season and is distinguished by the abundance of *Salix salviifolia* subsp. *australis* and the absence or scarcity of nemoral species in the shady understory, due to the substratum instability and strong sediment carriage.

VII—*Clematido flammulae-Salicetum australis ass. nova hoc loco* (clusters 14–18, Figure 1; *holotypus* relevé: 04, Table 4)

In the southern Portuguese territories included in the biogeographical unit of Algarve District, occurs the association *Clematido flammulae-Salicetum australis*, which is proposed here as a new willow forest association, exclusive from the limestone substrates of Barrocal algarvio, under thermomediterranean dry to subhumid belts. These riparian forests develop along banks of torrential Algarve's streams, where the hydrographical basin substrata are mostly basic [37]. It is

found on oligotrophic soils, with sandy-limey texture, that are periodically flooded, and resisting to prolonged drought periods. The new willow forest is dominated by *Salix salviifolia* subp. *australis*, constantly accompanied by *Salix neotricha* and *Clematis flammula*. The presence of Fraxinus angustifolia and *Nerium oleander*, reveals the catenal relationship of this association, with the ash woodlands (*Ranunculo ficariiformis-Fraxinetum angustifoliae*) and oleander micro-woodlands (*Oenantho crocatae-Nerietum oleandri*). Out of the trees and shrub observed, the climbing plant species are well represented, with *Aristolochia baetica, Lonicera implexa, Rubus ulmifolius, Smilax aspera* var. *altissima*, and *Calystegia sepium*. In the understory, we can highlight the presence of *Brachypodium sylvaticum, Vinca difformis, Equisetum ramosissimum,* and *Festuca ampla,* as well as hygrophilous plants from *Molinio caeruleae-Arrhenatheretea elatioris* and *Magnocarici elatae-Phragmitetea australis* such as *Oenanthe crocata, Carex hispida, Lythrum salicaria, Cyperus longus* subsp. *badius, Schoenoplectus lacustris, Typha domingensis, Scirpoides holoschoenus, Mentha suaveolens, Agrostis stolonifera, Dorycnium rectum*, among others, associated to environments affected by temporary flooding. The presence of nemoral-thermophile species *Aristolochia baetica, Bupleurum fruticosum*, and *Cheirolophus sempervirens*, must be emphasized.

**Table 4.** Clematido flammulae-Salicetum australis ass. nova hoc loco (Salicion salviifoliae, Salicetalia purpureae, Salici purpureae-Populetea nigrae).

| Ordinal Number | 1 | 2 | 3 | 4 | 5 |
|---|---|---|---|---|---|
| Surface (m$^2$) | 200 | 100 | 250 | 300 | 200 |
| Altitude (m) | 160 | 30 | 130 | 185 | 155 |
| Cover rate (%) | 85 | 85 | 95 | 90 | 85 |
| Orientation | NE | N | O | O | S |
| Slope (%) | 3 | 2 | 15 | 3 | 2 |
| Average height (m) | 6 | 6 | 6 | 7 | 6 |
| Number of species | 18 | 16 | 14 | 31 | 29 |
| Association characteristics | | | | | |
| *Salix salviifolia* subp. *australis* | 4 | 4 | 4 | 4 | 4 |
| *Salix neotricha* | 3 | 3 | 3 | 3 | 3 |
| *Clematis flammula* | + | + | + | 1 | + |
| *Vinca difformis* | . | 1 | 1 | 1 | 1 |
| *Equisetum ramosissimum* | + | 1 | . | + | 1 |
| *Fraxinus angustifolia* | . | . | + | 1 | + |
| *Brachypodium sylvaticum* | . | . | . | 1 | + |
| *Aristolochia paucinervis* | . | . | . | . | + |
| *Equisetum telmateia* | . | . | . | . | + |
| Companions | | | | | |
| *Rubus ulmifolius* | 1 | 2 | 1 | 1 | 2 |
| *Ceratonia siliqua* | + | + | 1 | + | + |
| *Smilax aspera* var. *altissima* | 1 | . | 2 | 1 | 2 |
| *Scirpoides holoschoenus* | + | + | . | 1 | 1 |
| *Nerium oleander* | 1 | . | + | 1 | + |
| *Arundo donax* | + | 1 | 1 | . | + |
| *Lythrum salicaria* | + | 1 | . | + | + |
| *Mentha suaveolens* | + | + | . | + | + |
| *Dorycnium rectum* | . | + | . | + | 2 |
| *Carex hispida* | + | . | . | + | + |
| *Rosa pouzinii* | . | 1 | . | + | . |
| *Calystegia sepium* | . | 1 | . | . | + |
| *Viburnum tinus* | + | . | . | + | . |
| *Aristolochia baetica* | . | + | . | . | + |
| *Paspalum dilatatum* | . | + | . | + | . |
| *Dittrichia viscosa* subsp. *revoluta* | . | . | + | . | + |

**Table 4.** *Cont.*

| Ordinal Number | 1 | 2 | 3 | 4 | 5 |
|---|---|---|---|---|---|
| *Samolus valerandi* | . | . | . | + | + |
| *Cyperus longus* subsp. *badius* | . | . | . | + | + |

Other taxa: + Euphorbia hirsuta, + Festuca ampla, + Oenanthe crocata, + Rosa canina in 1; + Lonicera implexa, + Arbutus unedo, + Rhamnus alaternus in 3; + Adiantum capillus-veneris, + Agrostis stolonifera, + Holcus lanatus, + Plantago major, + Prunella vulgaris, + Sanguisorba hybrida, + Rumex crispus, + Tamarix africana, + Typha domingensis, + Verbena officinalis, in 4; + Bupleurum fruticosum, + Dactylis hispanica subsp. glomerata, + Cheirolophus sempervirens, + Myrtus communis, + Schoenoplectus lacustris in 5. Localities (Coordinate Reference System Datum WGS84): 1—Fonte Filipe (lat 37°10′52.95″ N, long 7°57′46.71″ W); 2—Rib.ª da Fonte da Benémola (lat 37°12′32.69″ N, long 8°00′33.95″ W); 3—Amendoeiras (São Brás de Alportel e Estoi lat 37°07′24.21″ N, long 7°53′58.50″ W); 4—Rio Séqua (Asseca; lat 37°09′08.91″ N, long 7°40′21.63″ W); 5—Rib.ª da Fonte da Benémola (lat 37°11′50.11″ N, long 8°00′22.79″ W).

Its fringe and first substitution step belongs to the bramble shrublands of *Rubus ulmifolius* from *Lonicero hispanicae-Rubetum ulmifolii*. Accordingly, the removal of tree and shrub cover leads to the hygro-nitrophilous reed beds belong to the association *Holoschoeno vulgaris-Juncetum acuti*. Lastly occurs the perennial grasslands from *Narcisso willkommii-Festucetum amplae*, dominated by *Festuca ampla*, growing on neutro basic deep soils, with sandy-limey texture, along torrential streams running through calcareous deposits of the Jurassic and Cretaceous age, always in the Algarve's streams basin [38].

We place the Clematido flammulae-Salicetum australis ass. nova hoc loco, at the syntaxonomic level, in the Salicion salviifoliae alliance (Salicetalia purpureae, Salici purpureae-Populetea nigrae) and we propose changing its diagnosis to encompass both neutro-calcicolous and silicicolous communities.

## 4. Conclusions

With the information collected from 84 phytosociological relevés, we conducted a comparative and synthetic analysis of riparian woodlands communities dominated by *Salix atrocinerea* and *Salix salviifolia* subsp. *australis*, occurring in Southwestern part of the Iberian Peninsula and Sardinia. This study allowed us to identify the ecological position of *Salix atrocinerea* next to springs located hills sploes on psamophilic soils. Accordingly, we propose a new association namely *Frangulo baeticae-Salicetum atrocinereae*, for the acid pliocene-pleistocene sandy soils, in the Ribatejo and Sado Sector. Furthermore, for neutro-basophilous torrentail streams and watercourses of the Algarvese District (Algarve and Monchique biogeographic Sector) we propose the association *Clematido flammulae-Salicetum australis*.

At sub-serial level we identified a new perennial grassland association dominated by *Molinia caerulea* subsp. *arundinacae*, namely *Cheirolopho uliginosii-Molinietum arundinaceae*, for the thermomediterranean dry to sub-humid areas of the Ribatejo and Sado Sector, and represents a regression stage of *Frangulo baeticae-Salicetum atrocinereae*.

Such willow forests and perennial grasslands are important for biodiversity conservation, since they constitute a refuge to endemic or protected species, such as *Salix salviifolia* subsp. *australis*, *Euphorbia uliginosa, Cheirolophus uliginosus, Leuzea longifolia, Agrostis juressi, Euphorbia transtagana, Hyacinthoides transtagana*, among others. Therefore, it is important to guarantee the sustainability of these habitats, through conservation measures to avoid the disturbance on riparian woodlands across the Mediterranean, as reported by Fenu et al. [39]. In this sense, it would be necessary to create joint policies for these areas of high biodiversity, through the enhancement of ecosystem services.

Species conservation should give priority to places with a high number of plants threatened by IUCN (International Union for Conservation of Nature) criteria, such as Mediterranean waterways. Set as priorities, these areas should be subject to overgrazing control (mainly cattle grazing), develop programs to eradicate invasive plants (focusing on several species of the genus *Acacia* and *Arundo*), and seed harvest for germination and conservation studies in germplasm banks.

Attempt to their high conservational value, the willow woodlands associations should incorporate the habitat 92A0-*Salix alba* and *Populus alba* galleries, besides the perennial grasslands of *Cheirolopho*

*uliginosii-Molinietum arundinaceae*, incorporate the habitat 6410-*Molinia* meadows on calcareous, peaty or clayey-silt-laden soils (*Molinion caeruleae*), both from Annex I of Council Directive 92/43/EEC.

**Author Contributions:** Conceptualization, M.R. and R.Q.-C.; methodology, C.P.G.; software, M.R.; validation, C.P.G., G.S.; formal analysis, M.R., A.C.-O.; investigation, M.R., A.C.-O.; resources, M.R., R.Q.-C.; data curation, M.R., R.Q.-C., C.P.G.; writing—original draft preparation, M.R.; writing—review and editing, R.Q.-C., G.S.; visualization, A.C.-O.; supervision, C.P.G., G.S. All authors have read and agreed to the published version of the manuscript.

**Funding:** This research received no external funding.

**Acknowledgments:** With the contribution of the European Commission's LIFE program, through the Life-Relict Project—Preserving Continental Laurissilva Relicts (NAT/PT/000754).

**Conflicts of Interest:** The authors declare no conflict of interest.

## Appendix A. Syntaxonomical Scheme

*ALNETEA GLUTINOSAE* Br.-Bl. and Tüxen *ex* Westhoff, Dijk, and Passchier 1946

    *ALNETALIA GLUTINOSAE* Tüxen 1937

    *Alnion glutinosae* Malcuit 1929

        *Salici atrocinereae-Alnenion glutinosae* Rivas-Martínez, T.E. Díaz and F. Prieto 2011

          **Carici lusitanicae-Salicetum atrocinereae** Neto, Capelo, J.C. Costa and M. Lousã 1996

          **Frangulo baeticae-Salicetum atrocinerea** *ass. nova hoc loco*

          **Viti sylvestris-Salicetum atrocinereae** Rivas-Martínez and Costa *in* Rivas-Martínez, Costa, Castroviejo and E. Valdés 1980

*SALICI PURPUREAE-POPULETEA NIGRAE* (Rivas-Martínez and Cantó *ex* Rivas-Martínez, Báscones, T.E. Díaz, Fernández-González and Loidi 1991) Rivas-Martínez and Cantó 2002

    *POPULETALIA ALBAE* Br.-Bl. Ex Tchou 1948

   OSMUNDO-ALNION (Br.-Bl., P. Silva et Rozeira 1956) Dierschke et Rivas-Martínez in Rivas-Martínez 1975

        *Hyperico hircini-Alnenion glutinosae* Dierschke 1975

        **Carici microcarpae-Salicetum atrocinereae** Angius and Bacchetta 2009

        **Myrto communis-Salicetum atrocinereae** Biondi and Bagella 2005

   *SALICETALIA PURPUREAE* Moor 1958

  *Salicion salviifoliae* Rivas-Martínez, T.E. Díaz, F. Prieto, Loidi and Penas 1984

        **Clematido flammulae-Salicetum australis** *ass. nova hoc loco*

        **Salicetum atrocinereo-australis** J.C. Costa & Lousã *in* J.C. Costa, Lousã and Paes 1998

*MOLINIO-ARRHENATHERETEA* Tüxen 1937

   *HOLOSCHOENETALIA VULGARIS* Br.Bl. ex Tchou 1948

  *Molinio arundinacea-Holoschoenion vulgaris* Br.Bl. ex Tchou 1948

        *Brizo minoris-Holoschoenenion vulgaris* (Rivas Goday 1964) Rivas-Martínez in Rivas-Martínez, Costa, Castroviejo and E. Valdés 1980

          **Cheirolopho uliginosii-Molinietum arundinaceae** *ass. nova hoc loco*

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
