# Peer review of "Originalities of Willow of Salix atrocinerea Brot. in Mediterranean Europe"

_sustainability, doi:10.3390/su12198019_

Round 1
Reviewer 1 Report
Your study appears to be well conducted using accepted phytosociological methods. And, in this context, the manuscript is well written. However, it is unclear to me why you have submitted your paper to this journal. The term 'sustainability' - the journal title - is not mentioned anywhere. To be relevant for the readers of this journal you need to reframe the Introduction and better conceptualise the Discussion.
The English needs improvement. I have made a number of suggested edits in the attached pdf.
There is a need to better explain your methods - see comments in the pdf.
A map showing the location of the releves is needed with labels of the clusters.

Author Response
I appreciate your important suggestions.
Yes, this article is mainly about environmental sustainability, which in turn influences economic and social sustainability. These ideas were introduced in the introduction and conclusion.
The English suggestions were accepted and changed.
The methods were described in greater depth and point 2.1. Data collection was added.
Aa location map was also introduced where the relevés were carried out.
Thank you for helping to improve the article.
Reviewer 2 Report
I am not an expert of the Mediterranean vegetation, thus my review focuses ont he suitability of the methods and presentation of new results.
In my view, the most important result is that the authors recognized and described new associations. It is worth for publication. However, I have found several weaknesses in the present manuscript:
Line 47: Beyond the Spanish and Portugal syntaxonomical synthesis, you should refer to here the pan-European one complied by Mucina et al. and published in Applied Vegetation Science.
Line 74: Please explain the differential characteristics of the Sigmatists school within the Zürich-Monpellier approach. While the latter is widely known, this school is not.
Line 78: Euclidean distance is sensitive to dominant species, and down-weight species with lower cover, therefore its using in vegetation science is not recommended. It would be more convincing, if same clusters appeared in dendrograms created by other distance functions and clustering algorithms.
Line 82: Correctly: Rivas-Martinez et al. [28, 29]
Line 88: “plot 14-18” instead of “clusters 14-18” (this error repeated several places later)
Table 1: Beyond constancy values, fidelity of species should be shown. For calculation of fidelity see Chytry et al. (2002).
Table 2,3,4: “Number of species” instead of “Number of plants”
References:
Chytrý, M., Tichý, L., Holt, J., & Botta-Dukát, Z. (2002). Determination of diagnostic species with statistical fidelity measures. Journal of Vegetation Science, 13, 79–90. (V/16).
Author Response
I appreciate your important suggestions.
Yes, in order to contextualize the article at European level, the mucin reference has been added.
This is the same approach. However, it was known by some researchers as a sigmatist due to the station's name - SIGMA (International station for Mediterranean and Alpine geobotany). I rectified the meaning of the sentence by putting in parentheses.
Correct, we chose this type of analysis with Euclidean distance because we work with dominant species, similar to other works such as: https://doi.org/10.3390/plants9060741; https://doi.org/10.3390/su11041111; https://doi.org/10.1080/11263504.2012.761289
The remaining improvements were accepted and introduced directly to the text in blue.
Thank you for helping to improve the article.
Reviewer 3 Report
The manuscript entitled “Originalities of Willow of Salix atrocinerea Brot. in Europe Mediterranean” presents results of well-planned study on willow communities (genus Salix) in Mediterranean Europe. The introduction gives a good overview of the research problem and justifies the research. Field work was carried out in the southwest part of the Iberian Peninsula and in Sardinia through the analysis of 84 relevés (compiled according to the “Sigmatist” school of Zürich-Montpellier) submitted to hierarchical cluster analysis. The statistical methods used are well used and their results well described. The results are also well discussed with other similar studies. The conclusions do not raise any objections.
I only have one request for the authors because they highlight, starting from the title, the originality of these plant communities and the need to conserve them; however this aspect is only briefly mentioned in the conclusion section (LL278-279: “Such willow forests and perennial grasslands are important for biodiversity conservation, since they constitutes a refuges to endemic or protected species”) while in my opinion this topic deserves more attention (for a general conservation approach at Mediterranean level, the authors could refer to a recent paper published on “Diversity journal”: https://doi.org/10.3390/d12040157).
In line with this general issue, also the reference to plants protected by the Habitats Directive, endemic and endangered ones (L24) is resolved in only one general sentence in the conclusions (LL285-289). I agree that it is a relevant aspect and, for this reason, I think that this topic deserves a greater study by the authors.
Other minor points:
L20: change “living” by “growing”.
L24: delete “rare and/or”.
L26: “southwest of Iberian Peninsula”.
L27: use the italic font for the name of plants and phytocoenosis (please, revise all text since I found a lot of uncorrected use cases).
L29: change “endemism” by “endemic”.
L40: add adequate reference(s).
L58: “and in Sardinia”.
L61: “Costa et al.”.
L69: use the past tense (Biondi & Bagella described….).
L76: “Castroviejo et al. and Valdés et al.”.
LL76-77: “genus follows Portela-Pereira et al. [23].”
L78: add the reference of “the software R”.
LL81-82: for a detailed biogeographical characterization of Sardinia, please see Fenu et al. 2014 (https://doi.org/10.1080/14772000.2014.894592).
L84: “The cluster analyses (Figure 1) of 84 riparian…”.
L93: “La Maddalena archipelago (Sardinia)”; please revise also L70.
L104: delete comma (after studies).
L155: “Sinecologically, these willow woodlands inserted in…”.
L193: please, make reference to Fenu et al. 2014 for the detailed biogeographical characterization of Sardinia”.
LL205-209: the mentioned communities enriched with Laurus nobilis (classified as subass. lauretosum nobilis) are described in 2007 by Bacchetta et al. (and not by Biondi & Bagella 2005 as reported). Please correct this mistake and add the correct reference. I attach below the reference of this paper: Bacchetta G., Farris E., Fenu G., Filigheddu R., Mattana E., Mulè P., 2007. Contributo alla conoscenza dei boschi a Laurus nobilis L. della Sardegna, habitat prioritario ai sensi della Direttiva 92/43/CEE. Fitosociologia 44(2 - suppl. 1): 239-244.
L268: why unusual? please justify.
L272: “Algarve and Monchique biogeographic Sector”.
Author Response
I appreciate your important suggestions.
We add all your recommendations directly in the text.
Regarding the highlight of these communities, we add the importance of the high number of threatened plants through IUCN criteria and the main management actions that must be prioritized for a good conservation of the future.
Thank you for helping to improve the article.
Round 2
Reviewer 1 Report
The manuscript is improved with text placing the study into the context of sustainability. Grammatical issues remain, some of which I indicate below.
Line 44. What does 'sediment dimension' mean?
Line 49. "deepen the knowledge" would be better simply as "increase knowledge"
Line 50. What does "in order to contribute to adjust methods..." mean?
Line 54 "territory" is a political term?
Line 55. What is a "balanced landscape"?
Line 129. Why do you suggest that you wish to confirm new associations? I would think that you were trying to identify new associations.
Line 131. "...with according by proposed..." rephrase.
Line 146. clusters
Line 218. Omit "In regard to". Change "inserted" to "fit"
Author Response
Thank you very much for your suggestions. I think the article has improved significantly.
The corrections included in our article follow:
Line 44. What does 'sediment dimension' mean?
I was referring to the size of the soil sediments. So I changed it to: “size of substrate sediment”
Line 49. "deepen the knowledge" would be better simply as "increase knowledge"
Yes, I agree with your writing. Was changed.
Line 50. What does "in order to contribute to adjust methods..." mean?
Depending on the physiognomy of the land and the vegetation cover, it is necessary to select the most suitable methods. So we change “adjust” to “select”.
Line 54 "territory" is a political term?
Yes, we speak of territory as an administrative space.
Line 55. What is a "balanced landscape"?
Balanced landscape is a relative term, but it intends to convey the idea that the landscape is well-ordered, without major problems at the environmental, social or economic level.
Line 129. Why do you suggest that you wish to confirm new associations? I would think that you were trying to identify new associations.
Yes, to demonstrate the existence of new phytosociological associations, we collect data in the field, through inventories that undergo statistical analysis to confirm the results.
Line 131. "...with according by proposed..." rephrase.
We withdraw "by proposed"
Line 146. clusters
Introduced
Line 218. Omit "In regard to". Change "inserted" to "fit"
Corrections inserted.